

# Metabolite analysis reveals flavonoids accumulation during flower development in *Rhododendron pulchrum* sweet (Ericaceae)

Qiaofeng Yang[1,2,*], Zhiliang Li[3,*], Yuting Ma[3], Linchuan Fang[1], Yan Liu[3], Xinyu Zhu[3], Hongjin Dong[3] and Shuzhen Wang[3]

[1] Forestry and Fruit Tree Research Institute, Wuhan Academy of Agricultural Sciences, Wuhan, Hubei, China
[2] College of Food and Bioengineering, Henan University of Animal Husbandry and Economy, Zhengzhou, Henan, China
[3] Huanggang Normal University, Huanggang, China
* These authors contributed equally to this work.

## ABSTRACT

The azalea (*Rhododendron simsii Planch.*) is an important ornamental woody plant with various medicinal properties due to its phytochemical compositions and components. However little information on the metabolite variation during flower development in *Rhododendron* has been provided. In our study, a comparative analysis of the flavonoid profile was performed in *Rhododendron pulchrum* sweet at three stages of flower development, bud (stage 1), partially open flower (stage 2), and full bloom (stage 3). A total of 199 flavonoids, including flavone, flavonol, flavone C-glycosides, flavanone, anthocyanin, and isoflavone were identified. In hierarchical clustering analysis (HCA) and principal component analysis (PCA), the accumulation of flavonoids displayed a clear development stage variation. During flower development, 78 differential accumulated metabolites (DAMs) were identified, and most were enriched to higher levels at the full bloom stage. A total of 11 DAMs including flavone (chrysin, chrysoeriol O-glucuronic acid, and chrysoeriol O-hexosyl-O-pentoside), isoflavone (biochanin A), and flavonol (3,7-di-O-methyl quercetin and isorhamnetin) were significantly altered at three stages. In particular, 3,7-di-O-methyl quercetin was the top increased metabolite during flower development. Furthermore, integrative analyses of metabolomic and transcriptomic were conducted, revealing that the contents of isoflavone, biochanin A, glycitin, and prunetin were correlated with the expression of 2-hydroxyisoflavanone dehydratase (*HIDH*), which provide insight into the regulatory mechanism that controls isoflavone biosynthesis in *R. pulchrum*. This study will provide a new reference for increasing desired metabolites effectively by more accurate or appropriate genetic engineering strategies.

Corresponding author
Shuzhen Wang,
wszhen710@163.com

## INTRODUCTION

Evergreen azalea is an important ornamental woody plant in an environment where the temperature ranges from 20 °C to as low as −10 °C. The flower development of azaleas extended over two seasons. In the first season, the shoot apical meristems transformed into floral meristems and formed flower buds in August. From November to February, the flower buds were in a dormant state. After the dormancy, the development of flowers enters the partially open flower period in March, and then after the fully open flower period in April (*Cheon, Nakatsuka & Kobayashi, 2011*; *Cheon et al., 2012, 2013*). Anthocyanin compositions and components are the most important factors for various flower colors (*Tanaka, Sasaki & Ohmiya, 2008*; *He & Giusti, 2010*). There are many studies on the distribution of floral anthocyanins in *Rhododendron* (*Mizuta et al., 2009*; *Albert, Davies & Schwinn, 2014*; *Du et al., 2018*). Anthocyanin composition analysis of evergreen azaleas showed that the anthocyanin constitution of the purple group flowers is more varied than that of the red group flowers. Red series pigments are cyanidin and peonidin, and blue series pigments are delphinidin, petunidin, and malvidin (*Mizuta et al., 2009*). Moreover, flavonols containing the 3-hydroxy flavone backbone are also essential for flower colors (*Sheehan et al., 2016*; *Tan et al., 2019*). In azaleas, the red petal flavonol is quercetin and azaleatin, and the petals of the purple group had one to four flavonols: quercetin, azaleatin, myricetin and methyl-myricetin (*Mizuta et al., 2009*).

Except for the aesthetic value of the azalea flower, it has been used as a traditional Chinese medicine in China. The medicinal value of *Rhododendron* is perhaps due to the presence of various secondary metabolites. Plant secondary metabolites are crucial in regulating plant growth and development, and plant adaptive growth under biotic and abiotic stresses (*Chaves-Silva et al., 2018*; *Li et al., 2018*; *Xue et al., 2021*; *Zhan et al., 2022*). Generally, plant secondary metabolites include phenylpropanoids, terpenoids, and alkaloids. Flavonoids, a class of the most widespread phenylpropanoids in plants, are involved in many plant functions including pigmentation, plant reproduction, and protection against UV light and pathogens. The antioxidant, antimicrobial, and anticancer activity of flavonoids is becoming more attractive for humans (*Feng et al., 2016*; *Wang et al., 2016*; *Perez-Vizcaino & Fraga, 2018*; *Kopustinskiene et al., 2020*). Flavonoids are classified into six major subgroups: flavones, flavonols, flavanones, flavan-3-ols, anthocyanins, and isoflavones (*Routaboul et al., 2012*; *Dong et al., 2014*). Further modification reactions, such as glycosylation and acylation, raise the diversity of flavonoids (*Dong et al., 2014*; *Tan et al., 2019*).

Flavonoids are synthesized through the phenylpropane biosynthetic pathway. In this metabolic pathway, the precursors coumaryl-CoA and malonyl-CoA are catalyzed to naringenin sequentially by chalone synthesis (CHS) and chalcone isomerase (CHI). Naringenin is a key metabolite in the branch pathway for flavones, flavonols, isoflavone, and anthocyanins (*Nakatsuka et al., 2008*; *Tan et al., 2019*). In the flux of naringenin into the synthesis pathway for flavonols, flavonone-3-hydroxylase (F3H), flanonone-3′-hydroxylase (F3′H) and flavonol synthase (FLS) are key enzymes. In plants, myricetin, quercetin and kaempferol are the major flavonol form. It has been established that

flavonols biosynthesis was regulated by R2R3-MYB transcriptional factors including MYB11, MYB12, MYB21, and MYB111 (*Zhang et al., 2021*). Phytohormone also regulates flavonol biosynthesis. Recent work by *Shan et al. (2020)* revealed that gibberellic acid (GA) inhibits flavonol biosynthesis in Freesia hybrida. In Arabidopsis, auxin and ABA negatively correlated with root flavonol content. The ABA signaling pathway regulated flavonol biosynthesis, and in turn, flavonol may regulate the ABA signaling network (*Brunetti, Sebastiani & Tattini, 2019*). In addition, isoflavone is a distinct class among flavonoids and the isoflavone skeleton is derived from 2S-flavanones, such as naringenin and liquiritigenin. Isoflavones were mostly available in leguminous plants. Their biosynthesis has been proven to consist of two steps. The first step is catalyzed by a member of the CYP93C subfamily of cytochrome P450, 2-hydroxyisoflavanone synthase (IFS). The IFS product, 2-hydroxy-2,3-dihydrogenistein or 2,7,4′-trihydroxyisoflavanone is then dehydrated by 2-hydroxyisoflavanone dehydratase (HID) yielding daidzein or genistein in leguminous (*Akashi, Aoki & Ayabe, 2005*; *Du, Huang & Tang, 2010*; *Sohn et al., 2021*). This isoflavone participate in disease resistance and have a range of pharmaceutical and nutraceutical properties. For instance, daidzein is a common precursor to major phytoalexins, including medicarpin, biochanin A, and glyceollin (*Suzuki, Nishino & Nakayama, 2007*; *Wang, 2011*; *Szeja, Grynkiewicz & Rusin, 2017*).

As a result of population and universality in omics technologies, there are many studies on metabolomics analysis of different flower colors and transcriptomic analysis during flower development. For *Rosaceae*, flower flavonol, and anthocyanin distribution, phenolic content changes during flower development have been studied (*Schmitzer et al., 2010*; *Kanani et al., 2021*). For *Rhododendron* species, transcriptome analysis at different flower development stages and characterization of anthocyanins and flavonoids underlying flower color divergence have been conducted (*Du et al., 2018*; *Xia, Gong & Zhang, 2022*; *Ye et al., 2021*). *Rhododendron pulchrum* sweet is one of the most popular garden azalea cultivars and is a widely planted species in China, with attractive red purplish flowers. Transcriptomic analysis of flower development in R. *pulchrum* has been performed (*Yang et al., 2020*). To better understand the physiological, and biochemical processes that contribute to the visual changes underlying flower development, comprehensive flavonoid profiling of three flower development stages based on LC-MS were performed in this study. Additionally, integrated metabolomics and transcriptomics analysis were conducted. This research will provide valuable information to further elucidate the molecular mechanism of flower development in R. *pulchrum* and also provide an effective approach for the large-scale commercial production of health care value of azalea flowers.

## MATERIALS AND METHODS

### Plant material

The flowers of *Rhododendron pulchrum sweet* used in this study were harvested from Huanggang Botanical Garden in Hubei Province, China. The flowers of different developmental stages were respectively collected on March 20[th], April 6[th], and April 22[th], 2018 (*Wang et al., 2018*). The samples (at least six flowers of each sample) were

immediately frozen in liquid nitrogen and stored at −80 °C for subsequent analysis. There were two biological replicates per sample.

## Metabolite extraction

Samples of *R. pulchrum* flowers at three stages of development were selected for metabolomic analysis using ultra-performance liquid chromatography electrospray ionization tandem mass spectrometry (UPLC-ESI-MS/MS) system. The freeze-dried samples were ground to powder. Metabolites were extracted from 100 mg powder using 1.0 mL of 70% aqueous methanol solution overnight at 4 °C. The extracted solution was then filtered through a 0.22 µM microporous membrane and injected into LC-MS vials.

## Metabolic profiling of flavonoids

Exactly 5 µL of the sample were injected and analyzed using a Shim-pack UFLC SHIMADZU CBM30A system (Kyoto, Japan). Chromatographic separation conditions were as follows: column, Waters ACQUITY UPLC HSS T3 C18 column; the temperature of column oven, 40 °C; solvent system, A (0.04% formic): B (acetonitrile with 0.04% formic); flow rate, 0.4 mL/min. The eluting gradient program consisted of 0–11 min, 95–5% A; 11–12 min, 5% A; 12–12.1 min, 5–95% A; 12.1–15 min, 95% A. The QC sample was prepared by mixing aliquots of all the sample extracts and was injected after every set of 10 samples.

The effluent was connected to a triple quadrupole-linear ion trap mass spectrometer, the AB4500 Q TRAP system, and controlled by Analyst 1.6.3 software. The effluent was connected to a triple quadrupole-linear ion trap mass spectrometer, the AB4500 Q TRAP system, and controlled by Analyst 1.6.3 software. Metabolites qualitative and quantitative analysis followed the methods of *Chen et al. (2013)*. Based on the self-built database (Metware Biotechnology Co. Ltd., Wuhan, China), high-through quantification of metabolites was carried out by multiple reaction monitoring (MRM) for widely targeted metabolomics analysis. Based on the fragmentation pattern, retention time (RT), and mass-to-charge-ratio ($m/z$) values, metabolites in Metware's database were annotated by comparing with that of commercial standards or purified compounds, or searching the public databases, including MassBank, KNApSAcK, HMDB, MoTo DB, and METLIN. The QQQ scans were acquired as MRM experiments with the collision gas (nitrogen) set to 5 psi. Declustering potential (DP) and collision energy (CE) for each precursor-product ion (Q1–Q3) were done with further DP and CE optimization.

## Data analysis

To investigate the flower development-controlled accumulation of flavonoids, hierarchical clustering analysis (HCA), principal component analysis (PCA), and partial least squares-discriminate analysis (PLS-DA) were conducted. PCA and PLS-DA were performed using the SIMCA-P version 14.0 software. For HCA and PCA, the metabolite data were log2-transformed and followed by a min-max normalization. The heatmap was generated by the "heatmap.2" function in the "gplot" R-package. Differentially

accumulated metabolites (DAMs) were identified based on the thresholds of log2 (fold change) $\geq 1$ and variable importance for the projection (VIP) $\geq 1$ in PLS-DA. One-way ANOVA was used in this study ($p \leq 0.5$).

Transcriptome data were derived from previously published studies (*Wang et al., 2018*). We mapped the differentially expressed genes and DAMs simultaneously to the KEGG pathway database. Canonical correlation analysis (CCA) was carried out using the Vegan R package.

## RESULTS

### Flavonoid accumulation of *R. pulchrum*

Liquid chromatography tandem mass spectrometry (LC-MS) based metabolome analysis of *R. pulchrum* flowers at three stages (bud stage, partially open flower stage, and fully open flower stage), was performed to investigate the changes in flavonoid accumulation (Fig. 1). A total of 199 flavonoids, including 57 flavone, 42 flavonol, 32 flavone C-glycosides, 21 flavanone, and 12 isoflavone, were detected in our study (Table S1). To further investigate flavonoid accumulation in the different development stages, the flavonoid profile was statistically analyzed by hierarchical clustering analysis (HCA). Samples were divided into three groups according to the development stage, indicating that the accumulation of flavonoids displayed a clear development stage variation in terms of their abundance in different stages. In the fully open flower stage, almost half flavonoids reached the highest levels, followed by the bud and partially open flower stage (Fig. 2A).

We also analyzed the metabolite accumulation patterns by principal component analysis (PCA). The PCA results also showed a clear grouping of the metabolites into three distinct groups (Fig. 2B). The three main PCs (PC1, PC2, and PC3) accounted for 86% of the total system variability. The variables that also contributed to the PCs are listed in Table S2. The first component (PC1, 49.24%) separated the fully open flower stage from the bud and partially open flower stages, whereas the second component (PC2, 22.21%) separated the bud stage from the partially open flower and fully open flower stage, reflecting a major difference in metabolite levels among these three stages.

### Developmentally-controlled flavonoids in *R. pulchrum*

To determine the flavonoid accumulation patterns of *R. pulchrum* in flower development, a comparative metabolite analysis was performed. To identify flavonoids that mainly contribute to the separation of the different flower stages, partial least squares discriminant analysis (PLS-DA) was conducted, and the variable importance for the projection (VIP) values were used to identify the differentially altered metabolites (DAMs). Based on fold change (FC) $\geq 2$ or $\leq 0.5$ and VIP $\geq 1$, we identified a total of 25 DAMs between the bud and partially open flower stage. 63 DAMs between and fully open flower stage, and 60 DAMs between bud and fully open flower stage. Compared with the bud stage, there were 14 DAMs upregulated and 11 downregulated in the partially open flower stage. From the partially open flower stage to the fully open flower stage, 49 DAMs were upregulated and 11 DAMs were downregulated (Fig. 3A). The numbers of metabolites upregulated were comparable to or higher than the number of downregulated metabolites. Most
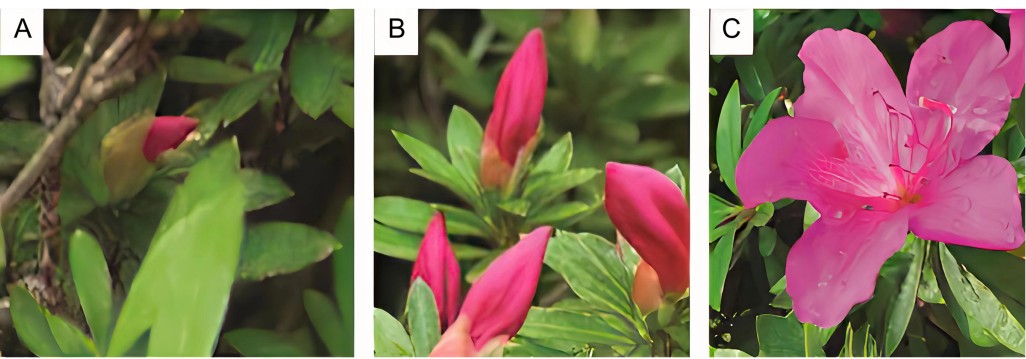

**Figure 1  Three flowering stages of *R. pulchrum*.** (A) Stage 1: bud; (B) stage 2: pre-flowering; (C) stage 3: fully open flower.

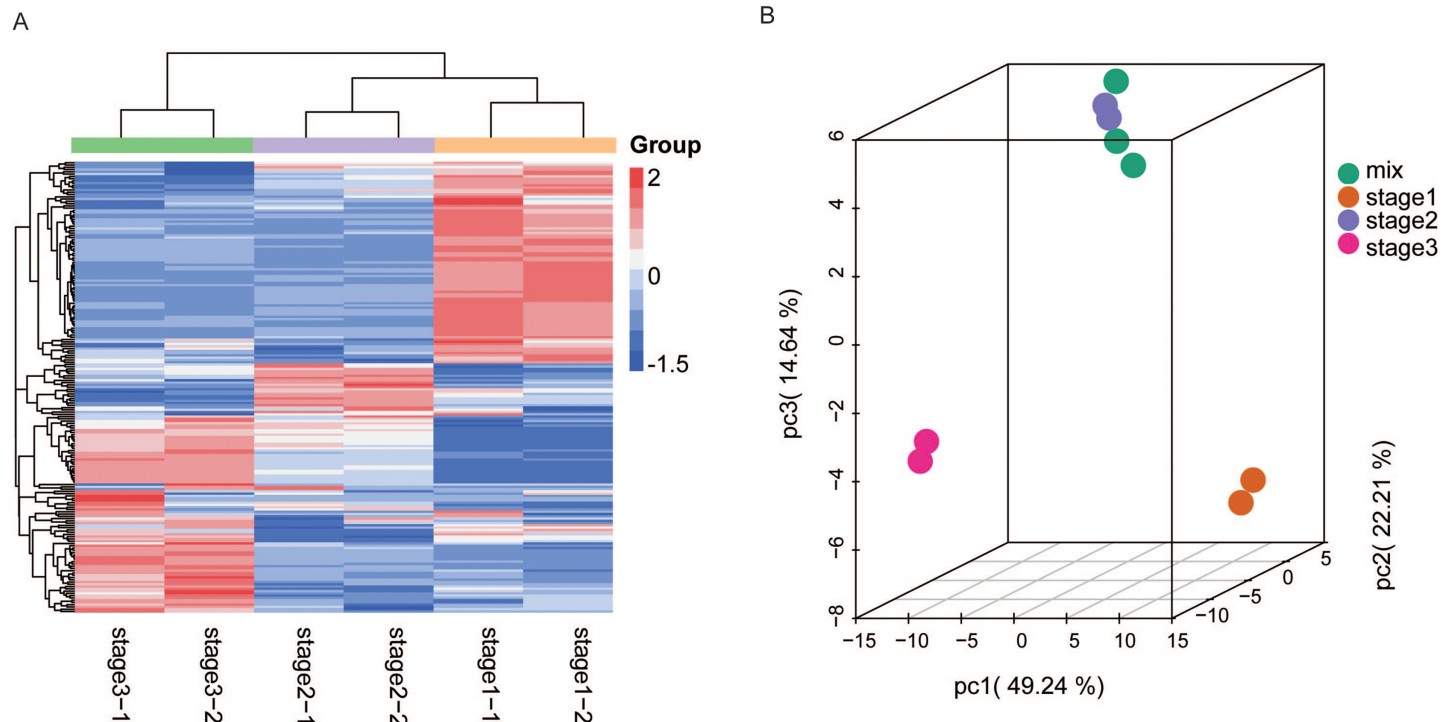

**Figure 2  HCA and PCA of flavonoids in *R. pulchrum* during flower development.** (A) Heatmap of flavonoids detected in the total samples. Red indicates high abundance, and green indicates low abundance. (B) Score plot of PCA in different flower development stages. Each point represents a sample from three stages and mixed samples. Stages 1, 2, and 3 represent the bud, pre-flowering, and fully open flower stages, respectively. The mix represents quality control (QC) sample, from the mixture of all samples.

anthocyanins, flavones, flavone C-glycosides, and flavonols were significantly higher at the fully open flower stage than the bud stage including delphinidin, pelargonin, chrysoeriol O-glucuronic acid, and kaempferol, indicating developmentally-controlled flavonoids accumulation in *R. pulchrum*.

Furthermore, the developmentally-controlled accumulation pattern of DAMs was compared among the three stages, and the results of clustering analyses showed that the

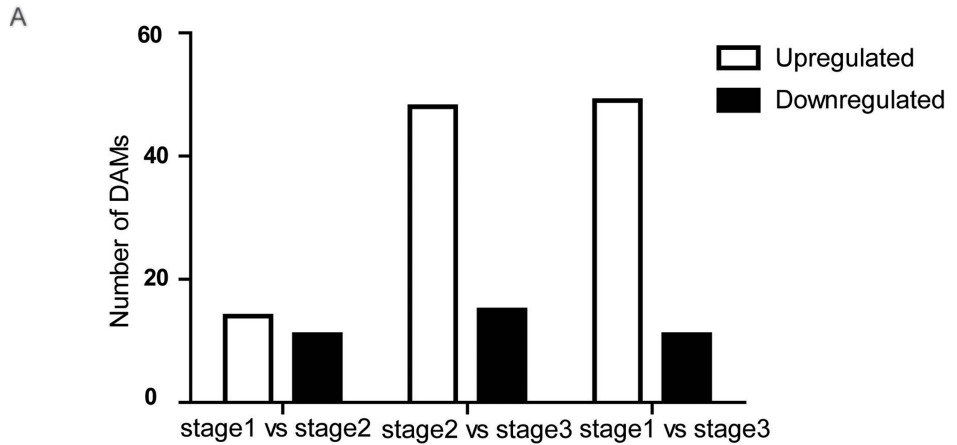

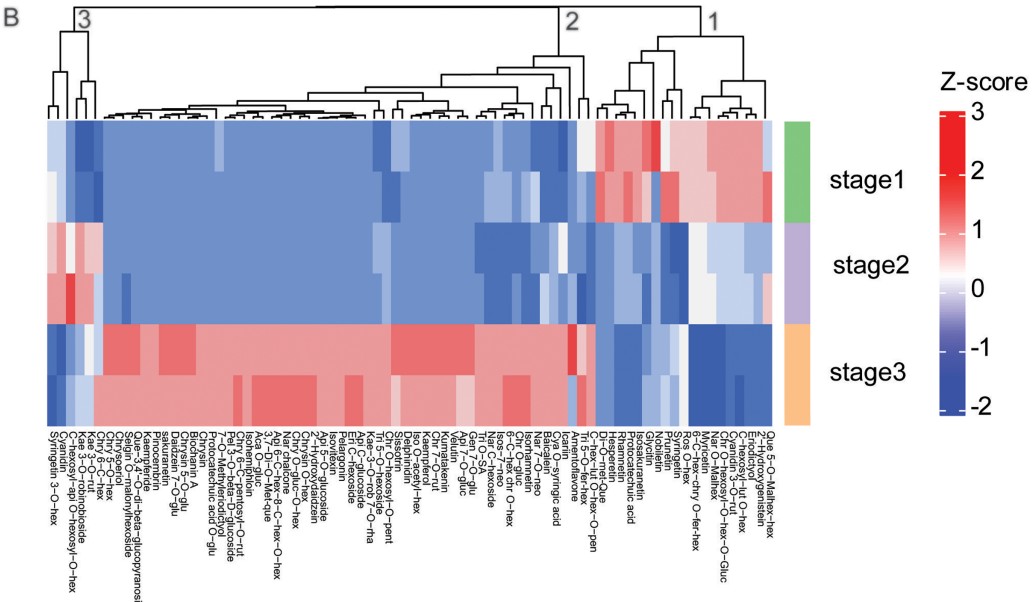

**Figure 3 Analysis of differential accumulation of metabolites (DAMs) during flower development.** (A) Numbers of upregulated or downregulated DAMs in the three comparisons. (B) Heat maps of 78 DAMs identified during flower development. Red indicates high abundance, and green indicates low abundance. All DAMs were divided into three clusters (1,2,3) according to their change trend. Ace, acetyl; Ade, adenosine; Api, apigenin; Aca, Acacetin; Chr, chrysoeriol; Cya, Cyanidin; GR, glucopyranoside; Dim, dimethyl; Eri, Eriodictyol; Fer, feruloyl; GE, b-guaiacylglyceryl ether; Glu, glucoside; Gly, glycerin; Gua, guanosine; Gluc, Glucuronic acid; Gen, Genistein; Hex, hexoside; hexosyl; Kae, Kaempferol; Ino, Inosine; Iso, Isorhamnetin; Isos, Isosakuranetin; Lut, luteolin; mal, malonyl; met, methyl; Nar, Naringenin; neo, neohesperidoside; Pen, pentoside;rut, rutinoside; oct, octadecatetraenoic acid; que, quercetin; Rob, robinoside; rha, rhamnoside; Ros, Rosinidin;Phe, phenylformic acid; Pen, pentosyl; Pel, Pelargonidin; Que, Quercetin; Sac, saccharopine; Sin, sinapoyl; SE, syringyl alcohol ether; syr, syringic acid; SA, saccharic acid; Tri,Tricin.

metabolites were grouped into three clusters (Fig. 3B). A deeper analysis of the metabolites in cluster 2 showed that anthocyanins, flavone, flavone C-glycoside, and isoflavone accumulated at higher levels at stage than at other two stages.

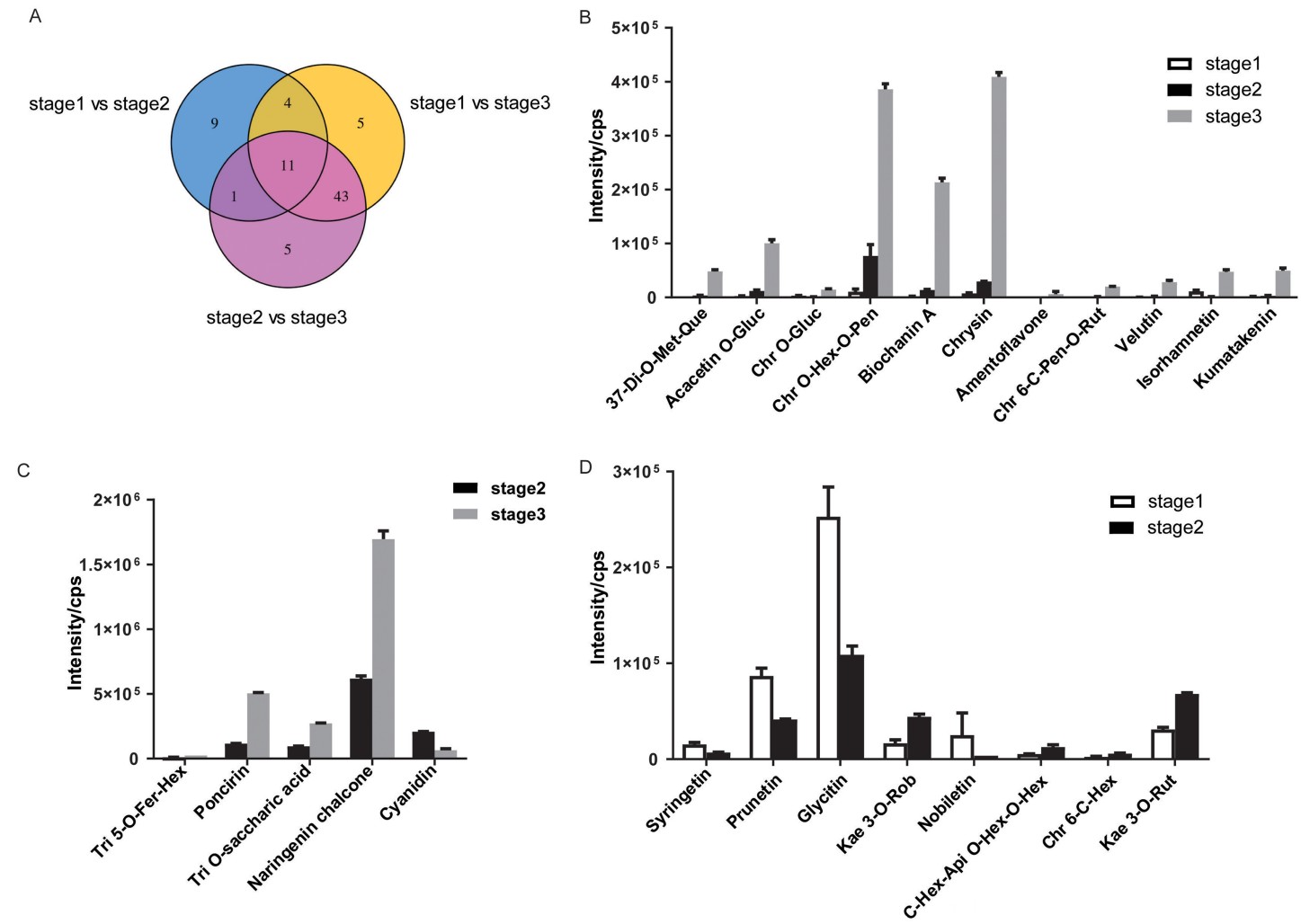

**Figure 4 Differential accumulation of metabolites (DAMs) in *R. pulchrum* during flower development.** (A) Venn diagram of DAMs shared by two comparisons or all three comparisons. (B) The 11 DAMs shared among the three comparisons. (C) five DAMs specifically exhibited a significant difference in accumulation between the pre-flowering (stage 2) and the fully open flower stage (stage 3). (D) The eight DAMs specifically exhibited a significant difference in accumulation between the bud (stage 1) and the pre-flowering stage (stage 2). The dentification of DAMs among different flowering stages was determined by PLSDA with the VIP values >1 and ANOVA ($p \leq 0.05$). The data in B, C, D are means ± SD from two biological replicates. Api, apigenin; Chr, chrysoeriol; Fer, feruloyl; Gluc, glucuronic acid; Hex, hexosyl; Kae, kaempferol; Met, methyl; Pen, pentoside; Que, quercetin; Rut, rutinoside; Ros, rosinidin; Tri, tricin.

To visualize the differences during flower development, the 78 DAMs were presented in Venn diagrams (Fig. 4A, Table S3). Several DAMs were shared between two comparisons: 12, 15, and 54 were shared between the bud/early and early/full, bud/early and early/full, and bud/early and bud/full comparisons, respectively. A total of 11 DAMs, including six flavones, three flavonols, one isoflavone, and one flavone C-glycoside, were in common during flower development. Among the 11 DAMs, the accumulation of niene metabolites significantly increased in *R. pulchrum* from bud to partially open flower to fully open flower stage and displayed the highest accumulation levels in the fully open flower stage, including 3,7-di-O-methyl quercetin, acacetin O-glucuronic acid, chrysoeriol O-hexosyl-O-pentoside, biochanin A, chrysin, chrysoeriol 6-C-pentosyl-O-rutinoside, velutin,

amentoflavone and kumatakenin (Fig. 4B). Chrysoeriol O-glucuronic acid and isorhamnetin were downregulated in the partially open flower stage and then upregulated in the fully open flower stage.

Differences in development stages may result in different accumulations of flavonoids. Nine metabolites specifically exhibited a significant difference accumulation between the bud and partially open flower stage, indicating that these metabolites may play roles in azalea from the bud to partially open flower stage (Fig. 4C). Syringetin, rosinidin O-hexoside, prunetin, glycitin, kaempferol 3-O-robinobioside (Biorobin), and nobiletin were down-regulated in the partially open flower stage, while C-hexosyl-apigenin O-hexosyl-O-hexoside, chrysoeriol 6-C-hexoside, and kaempferol 3-O-rutinoside (Nicotiflorin) were upregulated in the partially open flower stage.

During the flowering of *R. pulchrum*, five metabolites including tricin 5-O-feruloylhexoside, isosakuranetin-7-neohesperidoside (Poncirin), tricin O-saccharic acid, naringenin chalcone, and cyanidin, were characterized by a change exclusively in comparison between partially open flower and fully open flower stage, which may play roles in azalea from partially open flower stage to fully open flower (Fig. 4D).

## Differentially altered flavonol during flower development

Multiple functional roles of flavonoids in plant development have been reported. Flavonol is a major class of UV-absorbing compounds in plants, which are present in the upper epidermis of plant organs consistent with their role in UV protection. Copigmentation is an association between flavonols and anthocyanin pigments. The 78 DAMs identified in *R. pulchrum* during flowering included 17 flavonols, 23 flavones, 12 flavone C-glycoside, eight isoflavones, nine flavanones, and seven anthocyanins. From the bud to the partially open flower stage, eight flavonols were differentially accumulated. The content of 3,7-di-O-methyquercetin and chrysoeriol 6-C-pentosyl-O-rutinoside was 425 and 152-fold higher in the partially open flower stage, respectively (Fig. 5A).

During *R. pulchrum* flower development, significant differences were observed between the fully open flower stage and the other stage. The top ten elevated metabolites between the partially open flower stage and fully open flower stage included three flavonols: isorhamnetin, quercetin-3,4′-O-di-beta-glucopyranoside, and kumatakenin, which was 30, 21, 15-fold higher in the fully open flower stage. In addition, the top five reduced metabolites include three flavonols: quercetin 5-O-malonylhexosyl-hexoside, syringetin 3-O-hexoside, and myricetin (Fig. 5B).

## Identification of candidate enzymes involved in isoflavone biosynthesis during *R. pulchrum* flower development

To further explore the molecular mechanisms of flavonoid biosynthesis during flower development, integrative analyses of metabolomic and transcriptomic were conducted. The impacts of flower development on gene expression have been studied by comparative transcriptome before (*Wang et al., 2018*). The transcriptomic data were used to explore changes in flavonoids biosynthesis. We examined the expression of genes and the accumulation of metabolites during flower development. Based on the KEGG pathway

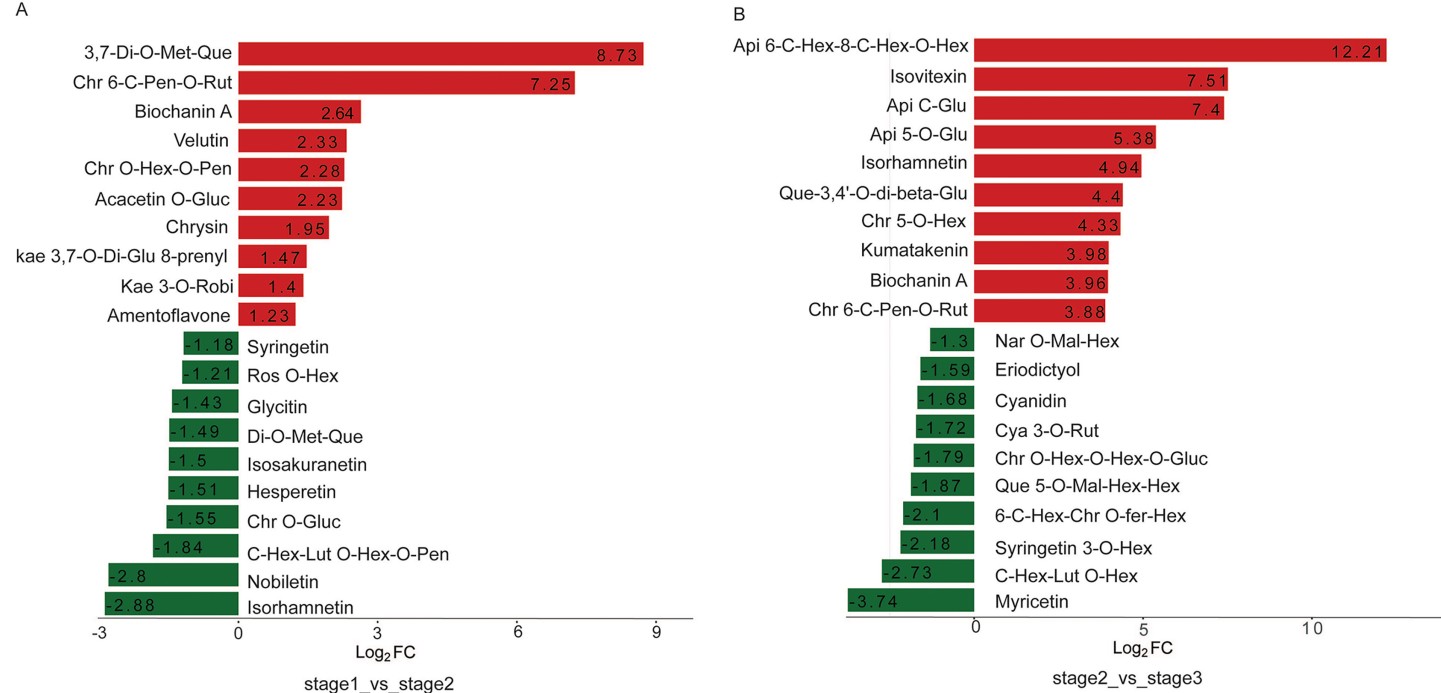

**Figure 5 Top 20 upregulated and downregulated DAMs in the two comparisons.** (A) Bud *vs* partially open flower stage, (B) partially open flower *vs* fully open flower stage. Api, apigenin; Chr, chrysoeriol; Cya, cyanidin; Fer, feruloyl; Glu, glucoside; Gluc, glucuronic acid; Hex, hexosyl; Kaem, kaempferol; Lut, luteolin; Mal, malonyl; Met, methyl; Nar, naringenin; Pen, pentoside; Que, quercetin; Rut, rutinoside; Ros, rosinidin; Robi, robinobioside.                                               

assignment, the isoflavone and flavonol biosynthesis pathway in *R. pulchrum* was constructed (Fig. 5). Significantly differential unigenes encoding putative enzymes involved in flavonoid biosynthesis have been shown. As illustrated in Fig. 6, transcription analysis showed that shikimate O-hydroxycinnamoyl transferase (*HCT*) and caffeoyl-CoA O-methyltransferase unigenes were down-regulated during flower development. While the expression level of naringenin 3-dioxygenase and flavonoid 3′-monooxygenase (*CYP75B1*) increased significantly during flower development. In addition, naringenin chalcone, biochanin A, kaempferol, and myricetin contents were highly accumulated in the fully open flower stage, whereas the contents of naringin, and 2-hydroxy genistein showed the highest level in the bud stage.

In particular, gene-metabolites networks analysis showed that the expression of *HIDH* (2-hydroxyisoflavanone dehydratase, K13258, TRINITY_DN75801_c0_g1) was 18-fold higher at the fully open flower stage than bud stage, a result consistent with the high content of biochanin A, while the production of hydroxylation modification of genistein, 2′-hydroxygenistein was significantly downregulated in the fully open flower stage. The expression of *HIDH* (TRINITY_DN66446_c1_g3) was five fold change at the partially open flower stage than bud stage, which is likely to participate in the higher accumulation of prunetin in the partially open flower stage (Fig. 6 and Fig. S1). These findings suggest that *HIDH* encodes function dehydratase, which is a critical determinant of isoflavone productivity during *R. pulchrum* flower development.

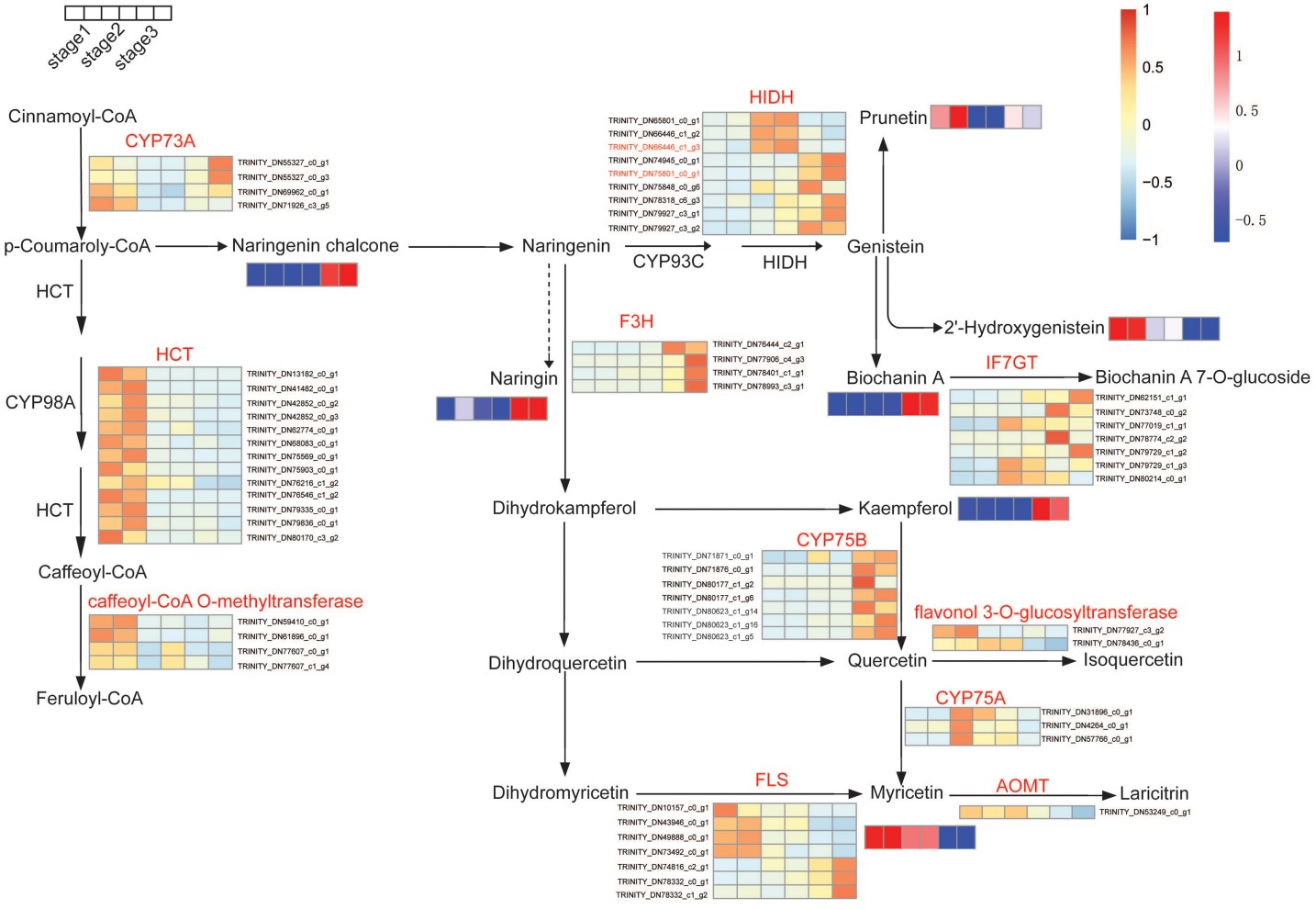

**Figure 6 Expression pattern of unigenes and accumulation profiles of DAMs in flavonoids biosynthesis pathway in *R. pulchrum* during flower development (three stages: bud, partially open flower, and fully open flower stage).** The color scale from blue (low) to orange (high) represents the FPKM values. The unigene names are indicated at the side of each step. The color scale from green (low) to red (high) represents the abundance of the metabolites. Unigene names were abbreviated as follows: trans-cinnamate 4-monooxygenase (CYP73A), shikimate O-hydroxycinnamoyl transferase (HCT), 2-hydroxyisoflavanone synthase (CYP93C), naringenin 3-dioxygenase (F3H), isoflavone 7-O-glucosyltransferase (IF7GT), flavonoid 3′,5′-hydroxylase (CYP75B), flavonol synthase (FLS), flavonoid 3′,5′-hydroxylase (CYP75A), flavonoid O-methyltransferase (AOMT).

## DISCUSSION

Using the widely targeted metabolomics method, a total of 199 flavonoids were detected in *R. pulchrum* flower. Coinciding with previous reports in evergreen azalea, cyanidin, delphinidin, quercetin, and their glycosides were the main types of anthocyanins or flavonols in *R. pulchrum* (*Du et al., 2018*). In contrast, pelargonidin (Pe), Pe 3-O-beta-D-glucoside, kaempferol, and its glycosides were newly detected in *R. pulchrum*. In the flowers of *Meconopsis*, flavonol glycosides such as kaempferol 3-O-glucoside, kaempferol 3-O-sophoroside also have been identified (*Yokoyama et al., 2018*). The presence of various flavonoid modifications suggested their important functions in flower developmental processes.

Flowers were considered as a strong sink of assimilates and the leaves continued to import dry matter to the corolla throughout its development until the final stage of senescence. During flower development, sustainable changes in the concentration of flavonoids happen. Research on the rose indicated that the petal accumulates total and specific phenolics until the full bloom stage (*Schmitzer et al., 2010*). In the present study, a similar accumulation pattern of flavonoids is established. The largest amount of flavonoids was also detected in the full flowering stage, such as kaempferol, myricetin, naringin, and delphinidin, which indicates that the full flowering stage is the best stage for obtaining those flavonoids to be used in pharmaceutical and therapeutic processes. Eridictyol, myricetin, prunetin and glycerin content were the highest at the bud stage. For rose, flavonoid content was also highest at the full open flower stage (*Kanani et al., 2021*). These results suggest that flower is a valuable source of flavonoids to promote human health. Based on developmentally dependent accumulation patterns of flavonoids, the different stage meets the demand for the special compound. To enhance the application value of azalea, harvesting the bud stage or the full flower stage is recommended.

Flavonol plays essential roles in plant growth, development, and communication with other organisms. Flavonol is also a major class of UV-absorbing compounds in plants, which absorbs light in the relevant range near-UV light (*Dudek et al., 2020*). Flavonol accumulation can be regulated by developmental stages and multiple environmental conditions such as light. Here, we found that nine flavonol contents increased from the bud to the fully open flower stage, such as 3,7-Di-O-methyquercetin, kaempferol 3-O-robinobioside, and kaempferol 3,7-O-diglucoside 8-prenyl (Fig. 4, Table S3). R2R3 MYBs and bZIP transcription factors are regulators of flavonol biosynthesis, such as *MYB11/111/12*, *LONG HYPOCCOTYL5* (*HY5*), and *HY5*-like *HYH* (*Bhatia et al., 2021*). Those transcription factors enhance the expression of flavonoids biosynthesis genes such as chalcone synthase (CHS), chalcone isomerase (CHI), flanonol synthase (FLS), and flavonoid 3'-hydroxylase (F3'H). HY5 is a central regulator of fundamental developmental processes such as seeding development, pigment accumulation, and abiotic stress response, which can regulate the transcription of numerous genes (*Gangappa & Botto, 2016*). In Arabidopsis, HY5 acts downstream of the constitutive photomorphogenesis 1 (COP1) for regulating flavonol accumulation in response to UV-B (*Bhatia et al., 2018*). Here, gene expression abundance during azalea flower development was quantified by calculating FPKM value, showing that *HY5* homologous was at least 3-fold up-regulated (Fig. S2). *F3H* expression level also increased during flower development. Further research on *HY5* and *F3H* will validate its function in flavonol biosynthesis during *R. pulchrum* flower development.

Isoflavones are the third largest family metabolites of the higher plants and are known for health-promoting phytoestrogenic functions. The 2-hydroxy isoflavones are early products of the isoflavonoids pathway and are then dehydrated by 2-hydroxyisoflavanone dehydratase (HIDH) to yield isoflavone including 4′-hydroxylated isoflavone and 4′-methoxylated isoflavone. In this study, 12 isoflavones were identified, including ten 4′-hydroxylated isoflavone and two 4′-methoxylated isoflavones (Table S1). Differential altered metabolites analysis showed that 2′-hydroxy daidzein, biochanin A, daidzin,

genistin, and sissotrin were significantly upregulated during *R. pulchrum* flower development. Especially, the content of biochanin A was 97-fold higher at the fully open flower stage than at the bud stage. We also analyzed the expression patterns of *HIDH* homologous during flower development. Furthermore, combined transcriptome and metabolite profiling revealed that *HIDH* expression levels were significantly correlated with biochanin A accumulation (Fig. S1), suggesting that HIDH regulates isoflavone biosynthesis in *R. pulchrum*. Similarly, HIDH has been proven to be a critical determinant of isoflavone productivity in hairy root cultures of Lotus japonicus (*Shimamura et al., 2007*).

HIDH proteins were members of a large carboxylesterase family which has distinct substrate specificity toward 4′-hydroxylated and 4′-methoxylated 2-hydroxyisoflavanones has been characterized. Kinetic studies revealed that *G. echinata* HIDH is specific to 2,7-dihydroxy-4′-methoxyisoflavanone, while soybean HIDH has broad substrate specificity toward 4′-hydroxylated and 4′-methoxylated 2-hydroxyisoflavanones (*Akashi, Aoki & Ayabe, 2005*; *Du, Huang & Tang, 2010*). In this study, though 4′-hydroxylated isoflavone was the main type identified in *R. pulchrum*, the two identified 4′-methoxylated isoflavones, biochanin A and sissotrin, were produced by the two steps: dehydration by HIDH and subsequent 4′-O-methylation catalyzed by HI4OMT. Therefore, further characterization of HIDH substrate specificity will provide insight into the regulatory mechanism that controls isoflavone biosynthesis in *R. pulchrum*.

# CONCLUSIONS

In this study, we analyzed the flavonoids profile during *R. pulchrum* flower development. A total of 199 flavonoids were detected, including 78 differential accumulated metabolites. Flavonoids displayed a developmentally controlled accumulation pattern, and most DAMs reached higher levels at the fully open flower stage. The flavonol, such as 3,7-di-o-methyquercetin content was differentially changed, suggesting that flavonol is an important factor during flower development. Additionally, through the combined analysis of the transcriptome and metabolomic data, we screened out the key enzyme HIDH which participates in isoflavone accumulation. The results of this study provide a deeper understanding of the molecular mechanism of flavonoid accumulation during *R. pulchrum* flower development.

## Funding

The research results reported in this article are funded by the National Natural Science Foundation of China (31500995), Natural Science Foundation of Henan Province (212300410151), Scientific and Technological Research Project of Hubei Provincial Department of Education (D20222902 and B2022204), as well as Hubei Provincial Key Laboratory of Economic Forest Germplasm Improvement and Comprehensive Utilization of Resources (202140804). The funders had no role in study design, data collection and analysis, decision to publish, or preparation of the manuscript.

## Grant Disclosures

The following grant information was disclosed by the authors:
National Natural Science Foundation of China: 31500995.
Natural Science Foundation of Henan Province: 212300410151.
Scientific and Technological Research Project of Hubei Provincial Department of Education: D20222902 and B2022204.
Hubei Provincial Key Laboratory of Economic Forest Germplasm Improvement and Comprehensive Utilization of Resources: 202140804.

## Competing Interests

The authors declare that they have no competing interests.

## Author Contributions

- Qiaofeng Yang conceived and designed the experiments, performed the experiments, prepared figures and/or tables, authored or reviewed drafts of the article, and approved the final draft.
- Zhiliang Li performed the experiments, prepared figures and/or tables, authored or reviewed drafts of the article, and approved the final draft.
- Yuting Ma analyzed the data, authored or reviewed drafts of the article, and approved the final draft.
- Linchuan Fang analyzed the data, prepared figures and/or tables, and approved the final draft.
- Yan Liu analyzed the data, authored or reviewed drafts of the article, and approved the final draft.
- Xinyu Zhu analyzed the data, authored or reviewed drafts of the article, and approved the final draft.
- Hongjin Dong analyzed the data, authored or reviewed drafts of the article, and approved the final draft.
- Shuzhen Wang conceived and designed the experiments, performed the experiments, authored or reviewed drafts of the article, and approved the final draft.

## Data Availability

The raw measurements are available in the Supplemental Files.

## Supplemental Information

Supplemental information for this article can be found online at http://dx.doi.org/10.7717/peerj.17325#supplemental-information.

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
