# Peer review of "Metabolite analysis reveals flavonoids accumulation during flower development in *Rhododendron pulchrum* sweet (Ericaceae)"

_PeerJ, doi:10.7717/peerj.17325_

## Round 0.1 · original submission · Major Revisions

Your manuscript was reviewed by three independent experts in the field. All the reviewers find the work interesting but raised multiple issues which need to be addressed properly. The reviewers provide detailed comments in their reviews and pointed out the areas where the manuscript needs to be improved. I also read the manuscript carefully and largely agree with the reviewers' comments.

·

Basic reporting

1. Sufficient literature coverage is required.

2. Article structure is ok. However, some information is repeated in materials and methods, results, and discussion. The initial part of the discussion sounded like an introduction section.

3. Thorough English language editing is required.

Experimental design

1. Three stages (bud, pre-flowering (partial flowering), and flowering) of flower development were included in the metabolite profiling using UPLC-MS/MS. It is not how many tissues were sampled at each stage (time-point); what is the number of replications? What is the variance (Analysis of Variance), one-way and two-way ANOVA? How many samples, including replication information, must be included in the materials and methods? How were the tissues collected?

2. Authors have used the previously published transcriptome data to compare the DAMs with DEGs. Were the transcriptome data generated for these three stages (at what conditions)? Include this information in the materials and methods section.

Validity of the findings

1. Though several DAMs were observed between different stages. I have seen overlapping or common metabolites between different stages. Discuss those common metabolites and explain the flux of different flavonoids over a period of flower development.

2. A further detailed discussion is required for the candidate gene identification part using the metabolite (DAMs) and expression data (DEGs).

Reviewer 2 ·

Basic reporting

The manuscript describes a study on the changes in flavonoids in three different stages of flower development in Rhododendron pulchrum. Data from the metabolomics were correlated to those from the transcriptomics (collected from previous study) to provide some insights into the biosynthesis of key flavonoids in this species. This research is interesting from the phytochemistry point of view. The manuscript is well-written, and data presentation and analysis are acceptable.

Comments:
1. Please replace Fig1A with one that has higher resolution.
2. Please explain what is represented by the error bars.

Experimental design

The experimental design is fine but additional details are required for some parts.

Line 120: Please name the botanist who authenticate the samples and mention if herbariums samples are available.

Line 124: Are the 6 flowers originated from a single plant or different plants?

Line 125: Two biological samples are insufficient in metabolomics. More samples are needed to account for the biological variation (as mentioned above).

Line 131: Why did the author choose 70% MeOH rather than other % of aqueous alcohol or acidified aqueous alcohols that are also commonly used? What is the most efficient solvent system to extract flavonoids?

Line 145: Please clarify how the identification was done, if comparison was done to an existing database, what are the parameters used? Some of the compounds could have been confirmed using commercially available standards. This part is crucial since as many as 199 compounds were claimed to be identified in this study.

Validity of the findings

I have some reservations about the number of biological replicates used and the manner by which the identification of the flavonoids were carried out. By convention, a higher number of biological replicates is desired and this increases the accuracy of the identified differentially expressed metabolites. In most metabolomics studies, three or more replicates are needed for each treatment, some suggested a minimum of 6 (Rodrigues et al. 2019) and biological replicates should be distinguished from technical replicates. Such information is missing from the manuscript.

References
Rodrigues, A.M.; Ribeiro-Barros, A.I.; António, C. Experimental Design and Sample Preparation in Forest Tree Metabolomics. Metabolites 2019, 9, 285. https://doi.org/10.3390/metabo9120285

Additional comments

This study is interesting and the findings are potentially important in expanding our understanding on the expression of genes responsible for key metabolites in the biosynthesis of flavonoids in plants. Having said that, the discussion section should incorporate more comparison of the findings in the current study with those in the literature. Particularly, one would be asking if there's any major differences between the data from Rhododendron pulchrum with other plant species.

·

Basic reporting

In manuscript “Metabolite and transcriptome analysis reveal flavonoids accumulation during flower development in Rhododendron pulchrum sweet “authors performed the comparative analysis of the flavonoid profile was in Rhododendron pulchrum sweet at three stages of flower development, bud (stage 1), partially open flower (stage 2), and full bloom (stage 3). A total of 199 flavonoids, including 78 differentially accumulated metabolites (DAMs) were identified during flower development, furthermore, integrative analyses of metabolomic and transcriptomic were conducted.

1. Authors have used transcriptome data from previously published study (Wang et al., 2018), it is better to incorporate the experimental condition in method section whether it is similar study at transcriptome level to present studies and include a comparison table of transcriptomics and metabolomics data. It is suggested that authors should remove the word transcriptome from the title of manuscript as they only compare their data with reported study.
2. Authors should mention the botanical name of Azalea (Rhododendron pulchrum) at first place in abstract and Introduction section.
3. Fig 2A. HCA why author included stages 1-1 and 2-2 comparison? why not 3-3 included. In Fig.2B author should write in legend about mix, the green dot in figure.
4. Fig 3B. Author should and incorporate the DAMs name in the Heat map
5. Fig 4. Legend has unwanted characters ( <!--[if !supportLists]-->(A) <!--[endif]-->) delete it.
Supplemental data
Raw_data: sheet 1and 2 heading what is 3-1 in “Stage-2_vs_Stage-3-1” and “Stage-1_vs_Stage-3-1”? and what is Zihe? in Zihe-bud-stage-1, authors should check it carefully and rectify it.

Experimental design

no comment

Validity of the findings

no comment

---

## Round 0.2 · Major Revisions

The comments raised by the reviewers are not addressed by the authors satisfactorily. Therefore manuscript still needs revision(s) before acceptance in PeerJ. I would suggest authors address the queries raised by reviewer 2 during this as well as the previous review.

Reviewer 2 ·

Basic reporting

The authors attempted to improve the manuscript; however, some of my queries have not been fully addressed. Notably, some of the important chemistry data have not been included/clarified.

1. Did the authors deposit the specimen of the plant materials used in this study in any herbarium (a standard practice in this field)?
2. The authors stated that 70% MeOH was the best solvent system for extracting flavonoids, yet it is not clear what do they mean by "best" and whether this is deduced from the literature (on plants?) or their previous research. Please explain and incorporate this in the revised manuscript.
3. The authors clarified that several parameters including the m/z values of precursor ions and fragment ions generated from authentic standards were used in the metabolite identification process, yet, in Table S1, such information is missing. Without the MS/MS fragments, one is curious how the identification of the compounds was achieved. The author claimed that public databases (line 150) were used for comparison but there is no information about this in the manuscript. Data for the MS/MS fragments must be presented to support the identification of the metabolites.
4. Please include the PDA and TIC chromatograms in the revised manuscript.
5. The lack of biological replicates is indeed a limitation of this study, and how it may influence the findings of this study could have been stated.

Experimental design

None

Validity of the findings

None

Additional comments

None

·

Basic reporting

no comment

Experimental design

no comment

Validity of the findings

no comment

---

## Round 0.3 · Major Revisions

Authors partially addressed the comments raised by reviewer 2 who has valid concerns. E.g. Disclosing the information of metabolite extraction, detection, and analysis is critical for reproducibility. Reviewer 2 mentions that a direct comparison of the molecular weight based on the pseudomolecular ion in the LCMS can be erroneous and there is no proper explanation of how comparison to the long list of databases was made.

Reviewer 2 ·

Basic reporting

The authors have responded to some of the previous comments and amended the manuscript accordingly.

The authors added a list of public databases used in for metabolite identification in the methodology section, yet it is still unclear which compound was identified using information from which database (such information was not included in Table S1).

Table S1 contains essentially the same data as in the previous version except with a change in some of the column titles, namely the "precursor ion" and "fragments". In the "fragments" column, there is only one single mass - what do the authors mean by "main fragment"? The one with highest intensity/abundance? How do the authors identify the metabolites based on just one main fragment? MS/MS fragments should have been used for achieve more convincing identification.

It is unclear if MS/MS fragmentation was performed during the analysis as details such as collision energy was not provided.

If there is no MS/MS data, it is unclear how identification of the compounds was achieved by merely comparing the pseudomolecular ion and considering the fact that many organic compounds have the same molecular weight.

This paper reported that 199 flavonoids (a rather large number) were identified but the manner by which such compounds were identified is only vaguely presented. In my opinion, this is unacceptable for a scientific publication.

Experimental design

None

Validity of the findings

Accurate identification of compounds from LCMS analysis is crucial as it affects the interpretation of the results from the metabolomics platform.

Additional comments

The authors should provide detailed explanation regarding the analysis of the extracts and how the metabolites were identified, instead of a general version.

---

## Round 0.4 · Minor Revisions

It is a requirement of the journal that you share the raw data as pointed out by the reviewer.

Reviewer 2 ·

Basic reporting

The authors have responded to my comments and explained how the metabolite identification was carried out, however, the data cannot be shared due to database confidentiality (Author's rebuttal letter, page 5).
I appreciate the authors' effort to provide clarification.

Experimental design

No comment

Validity of the findings

No comment

Additional comments

Findings from this work improves our understanding on the relationship between metabolite concentration with developmental stages in the flower, using Rhododendron pulchrum as an example.

---

## Round 0.5 · accepted · Accept

Authors have addressed all the comments raised during the review process. The manuscript is ready for the publication.